# Mosaic patterns of selection in genomic regions associated with diverse human traits

**Abin Abraham**[1☯¤], **Abigail L. LaBella**[2,3,4,5☯], **John A. Capra**[6,7]*, **Antonis Rokas**[2,3,8,9]*

**1** Vanderbilt University Medical Center, Vanderbilt University, Nashville, Tennessee, United States of America, **2** Department of Biological Sciences, Vanderbilt University, Nashville, Tennessee, United States of America, **3** Evolutionary Studies Initiative, Vanderbilt University, Nashville, Tennessee, United States of America, **4** Department of Bioinformatics and Genomics, University of North Carolina at Charlotte, North Carolina, United States of America, **5** North Carolina Research Center, Kannapolis, North Carolina, United States of America, **6** Bakar Computational Health Sciences Institute, University of California, San Francisco, California, United States of America, **7** Department of Epidemiology and Biostatistics, University of California, San Francisco, California, United States of America, **8** Department of Biomedical Informatics, Vanderbilt University School of Medicine, Nashville, Tennessee, United States of America, **9** Vanderbilt Genetics Institute, Vanderbilt University, Nashville, Tennessee, United States of America

☯ These authors contributed equally to this work.
¤ Current address: Children's Hospital of Philadelphia, Philadelphia, PA, United States of America
* tony@capralab.org (J. A. C.); antonis.rokas@Vanderbilt.Edu (A. R.)

**Data Availability Statement:** Data Availability We have made both the formatted input files and the final output files (both trait and region level results) available for download on FigShare (the public link will be provided upon publication; for the purpose

## Abstract

Natural selection shapes the genetic architecture of many human traits. However, the prevalence of different modes of selection on genomic regions associated with variation in traits remains poorly understood. To address this, we developed an efficient computational framework to calculate positive and negative enrichment of different evolutionary measures among regions associated with complex traits. We applied the framework to summary statistics from >900 genome-wide association studies (GWASs) and 11 evolutionary measures of sequence constraint, population differentiation, and allele age while accounting for linkage disequilibrium, allele frequency, and other potential confounders. We demonstrate that this framework yields consistent results across GWASs with variable sample sizes, numbers of trait-associated SNPs, and analytical approaches. The resulting evolutionary atlas maps diverse signatures of selection on genomic regions associated with complex human traits on an unprecedented scale. We detected positive enrichment for sequence conservation among trait-associated regions for the majority of traits (>77% of 290 high power GWASs), which included reproductive traits. Many traits also exhibited substantial positive enrichment for population differentiation, especially among hair, skin, and pigmentation traits. In contrast, we detected widespread negative enrichment for signatures of balancing selection (51% of GWASs) and absence of enrichment for evolutionary signals in regions associated with late-onset Alzheimer's disease. These results support a pervasive role for negative selection on regions of the human genome that contribute to variation in complex traits, but also demonstrate that diverse modes of evolution are likely to have shaped trait-associated loci. This atlas of evolutionary signatures across the diversity of available GWASs will enable exploration of the relationship between the genetic architecture and evolutionary processes in the human genome.

of peer review, editors and reviewers can access the data via the private link https://figshare.com/s/7a995a6d08847d7b245f). The FigShare repository contains one compressed folder per PubMed ID, which contains all the associated input and output files. Code Availability Evolutionary calculations were performed using the GSEL python package, which is freely available at https://github.com/aa-publications/gsel_vec. Scripts with necessary data to replicate manuscripts will be provided upon publication.

**Funding:** This work was supported by the National Institutes of Health grants R35GM127087 (J.A.C.), R01HD101669 (A.R.), R56AI146096 (A.R.), and R01AI153356 (A.R.), the Burroughs Wellcome Fund Preterm Birth Initiative (J.A.C and A.R.), the National Science Foundation grant DEB-2110404 (A.R.), and by the March of Dimes through the March of Dimes Prematurity Research Center Ohio Collaborative (J.A.C. and A.R.). A.A. was supported by the American Heart Association fellowship 20PRE35080073 and NIGMS of the National Institutes of Health under award number T32GM007347. The funders had no role in study design, data collection and analysis, decision to publish, or preparation of the manuscript.

**Competing interests:** I have read the journal's policy and the authors of this manuscript have the following competing interests: A. R. is a scientific consultant for LifeMine Therapeutics, Inc.

## Author summary

Understanding how evolutionary forces shape patterns of human genomic variation is fundamental for evolutionary genomics and medicine. We developed a novel and very robust computational framework that measures enrichment for evolutionary forces acting on regions associated with variation in diverse complex traits. Application of this framework to more than 900 genome-wide association studies and 11 evolutionary measures, while accounting for potential confounders, generated a comprehensive evolutionary atlas that maps diverse signatures of selection on genomic regions associated with hundreds of complex human traits. Notably, genomic regions associated with human complex traits have been shaped by diverse modes of selection. Combined with the availability of a computational package that can perform these calculations for any set of genomic regions associated with any trait in any organism, this work is a major step forward toward understanding the relationship between genetic architecture and selection.

## Introduction

Understanding how natural selection has shaped the human genome is fundamental for evolutionary genomics and medicine [1]. As humans expanded out of Africa, they encountered diverse climates, underwent dietary changes, experienced demographic shifts, and mixed with Neanderthals and other hominins. The selective pressures exerted by these events and non-selective forces such as admixture, population demography, and genetic drift shaped the genetic basis of modern human traits [2–5]. Two well-known examples include the strong positive selection on adult milk consumption that shaped frequencies of lactase persistence alleles [6–8] and a Denisovan introgressed haplotype that contributed to high-altitude adaptation of Tibetans [9,10]. Although the evolutionary histories of these and several other specific loci and traits have been studied [11–14], the extent and types of evolutionary forces that have acted on the genomic regions associated with variation in the human phenome remain poorly understood.

Multiple measures have been developed to infer evolutionary forces from patterns of genetic variation within and between species [15]. For example, comparing human genomes to those of related species using measures like PhyloP and PhastCons enables testing hypotheses about decreases and increases in the substitution rate over evolutionary time. Decreased substitution rates are often indicative of the action of negative selection [16,17]. Identification of clusters of variants at intermediate allele frequencies in human populations by measures such as the Beta Score suggests balancing selection [18,19]. Similarly, measures such as $F_{ST}$ and XP-EHH rely on single nucleotide polymorphism (SNP) and haplotype structures to infer potential local adaptation or recent positive selection between human populations [20]. It is also possible to estimate the time to the most recent common ancestor of different haplotypes and quantify the age/origin of variants using ancestral recombination graphs [21]. Driven by increasing amounts of whole genome sequence data and computational power, more recent methods, such as RELATE [22] and CLUES [23], use locally constructed genealogies and ancestral recombination graphs to reconstruct allele histories and infer the action of recent directional selection. Other methods rely on parametric models of neutral evolution [24] or analyze patterns of singleton variants [25] that incorporate population level genomic data and GWAS summary statistics to estimate the strength of selection and evidence for directional selection [13,26]. Multiple different evolutionary forces can influence the values of the

evolutionary measures we consider; however, together they enable exploration of evidence for a diverse set of evolutionary forces from patterns of genetic variation.

Despite advances in these methods, which mainly focus on individual regions, mapping the evolutionary pressures on complex traits remains challenging for several reasons. First, genomic attributes that influence ascertainment and power in association studies, e.g., allele frequency and linkage disequilibrium (LD), also influence the expected distribution of many evolutionary metrics. Thus, a genomic background derived from averaging across all variants in the human genome does not provide an appropriate null when interpreting overlaps between trait associations and signatures of selection. Second, population stratification is common in genome-wide association studies (GWASs). As GWASs became more prevalent and demonstrated that most common traits are polygenic, new trait-focused approaches to detect evidence of recent polygenic selection emerged. Polygenic scores, which can be derived by summing across trait-associated alleles from a GWAS after weighting by the effect size, enable prediction of phenotype from genotype. Several studies computed polygenic scores across populations and interpreted systematic differences and the alleles that drive them as evidence of polygenic adaptation [27–29]. For example, human height increasing alleles identified from GWAS were found to be at consistently higher frequencies in Northern European populations compared to Southern Europeans [29]. However, subsequent analyses revealed that residual population stratification in the GWASs and a resulting lack of consistent effects across populations drove the initial signatures of selection [30–34]. Detecting and correcting for residual stratification is an ongoing challenge in the field. Nevertheless, certain evolutionary patterns have consistently emerged across many studies with varied approaches. Regions of the genome that have been associated with complex traits, such as hair color, body mass index, waist-to-hip ratio etc., consistently show evidence of recent and directional selection [13,22,35,36].

In this study, we describe a unified approach to measure enrichment for evolutionary forces acting on regions associated with variation in diverse complex traits. This approach is complementary to previous work on polygenic adaptation [24,35,37] because we characterize the evolutionary attributes of the genomic regions that contribute to complex trait variation. To protect against biases from stratification, our approach: 1) does not directly incorporate effect sizes at trait-associated regions (e.g. as in polygenic scores), 2) builds a null distribution from allele frequency and LD-matched SNPs, and 3) enables flexible enrichment testing at different association thresholds. We generate an atlas of 11 evolutionary measures on regions identified from GWASs of over 900 polygenic traits (totaling 210,109 genomic regions). We find widespread positive enrichment for sequence constraint, a dearth (i.e., negative enrichment) of patterns associated with balancing selection, and several groups of GWASs that show distinct positive enrichments for population differentiation. By mapping the evolutionary landscape of genomic regions that underlie specific complex traits, these results reveal that human trait-associated regions have been shaped by a mosaic of different modes of selection.

## Results

### An efficient permutation-based approach to infer evolutionary forces on GWAS loci

To explore genomic signatures of diverse evolutionary forces on genomic regions associated with complex human traits, we developed an empirical framework that infers enrichment for diverse evolutionary measures from GWAS summary statistics. For a given GWAS, we consider independent trait-associated genomic regions accounting for LD (r2>0.9, GWAS p-value < 5e-8, Fig 1A).

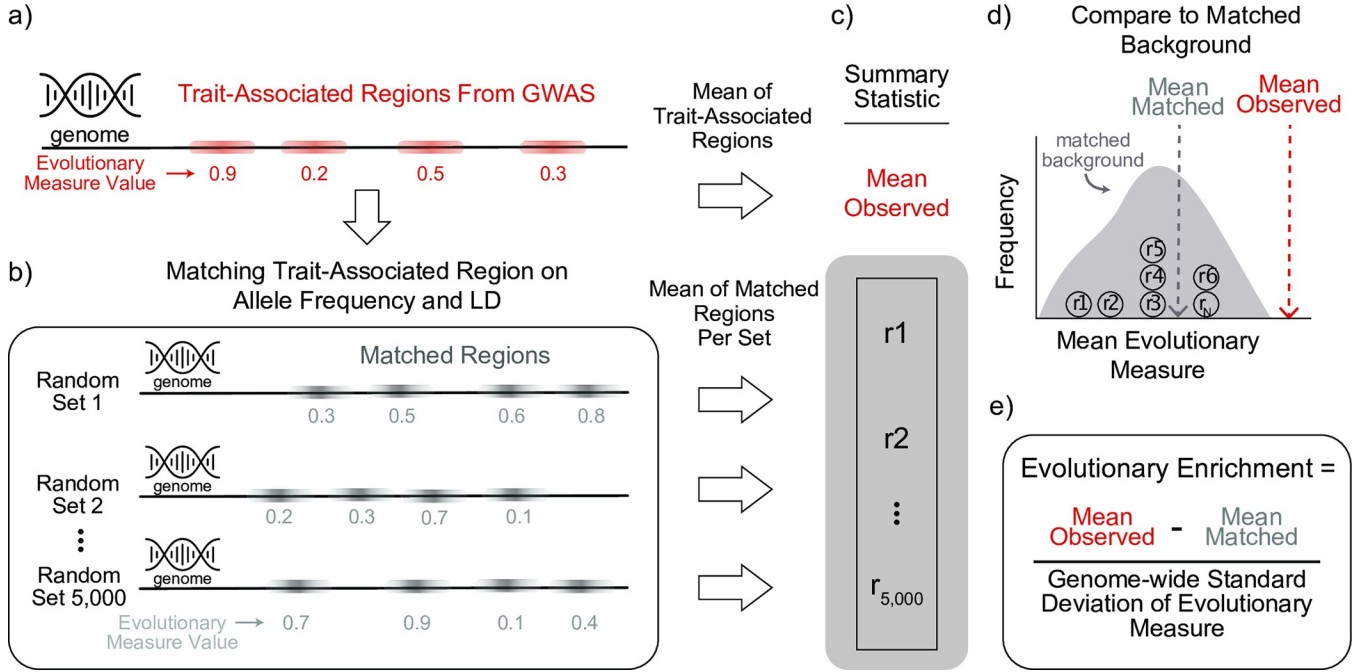

**Fig 1. Computational framework for detecting positive and negative enrichment for evolutionary signatures on genome-wide association studies (GWASs).** (a) Given the GWAS of a complex trait, we define trait-associated regions by first identifying variants of genome-wide significance and then clumping based on linkage disequilibrium (LD; e.g., r2>0.9). For each region, we identify the maximum value of an evolutionary measure of interest. (b) For each trait-associated region, we identify 5,000 randomly selected genomic regions ("matched regions") that have similar minor allele frequency, linkage disequilibrium, and gene proximity patterns (Methods). (c) Across the trait-associated regions and their matched random genomic regions, we calculate a summary statistic. To illustrate our approach, we take the mean of the evolutionary measure to generate an (d) empirical background distribution and (e) calculate enrichment by comparing the mean observed evolutionary measure to the mean of the matched background distribution. We divide by the standard deviation of the evolutionary measure across the genome to standardize the enrichment. However, any summary statistic of interest could be used.

To define an appropriate background distribution for each analysis, we randomly select genomic regions matched on minor allele frequency, LD patterns, and gene proximity for each trait-associated region. The matching is repeated until we have 5,000 sets that each contain the same number of genomic regions as the trait-associated regions (Fig 1B). For each evolutionary measure, we build a background distribution from each matched set. We then compare the observed trait-level evolutionary values to the background distribution and calculate an empirical p-value (Fig 1C and 1D). To summarize each comparison, we define the standardized evolutionary enrichment as the difference between the observed trait-level mean and the matched-background mean divided by the genome-wide standard deviation for the evolutionary measure (Fig 1E). Thus, the evolutionary enrichment may be higher (positive enrichment) or lower (negative enrichment) than the expected matched-background mean. However, we note that other statistics could be used to compare the observed and expected values.

We apply this approach for 11 evolutionary measures that quantify different patterns of genomic variation influenced by the action of different modes of selection, such as directional selection, balancing selection, local adaptation, and negative selection. All evolutionary measures had high coverage (83–99%) across the set of SNPs used in our study (Methods and Table 1).

## Evolutionary signals are consistent across multiple GWASs for height

To evaluate the robustness of our computational framework against potential differences in GWAS size, population, study design, and analysis strategy, we compared four GWASs

**Table 1. Evolutionary measures used to quantify trait-associated regions.** For each evolutionary measure (rows), the sequence pattern, type of evolutionary force suggested, and the corresponding time scale is given. Multiple selective and non-selective forces can shape these patterns. "%SNPs covered" is the proportion of SNPs from 1000 Genomes Phase III after quality control (n = 9,535,059) that have an annotation for the given evolutionary measure. For $F_{ST}$ and XP-EHH, we used the following 1000 genomes superpopulation comparisons: afr-eas, afr-eur, eas-eur. XP-EHH: cross-population extended haplotype homozygosity (EHH). TMRCA: time to most recent common ancestor derived from ARGweaver.

| Evolutionary Measure | Type of Evolutionary Signature | Suggested Evolutionary Force | Time Scale (in years) | %SNPs covered |
|---|---|---|---|---|
| ARGweaver | Time to Most Recent Common Ancestor (TMRCA) | N/A | Human population (~100 million years) | 99% |
| Beta Score | Clusters of alleles at similar intermediate frequency | Balancing Selection | Human Population (>10,000 of years) | 99% |
| PhyloP | Non-neutral substitution rates | Positive/Negative Selection | Across species (~100 million years) | 98% |
| PhastCons | Clustered low substitution rates | Negative Selection | Across species (~100 million years) | 98% |
| LINSIGHT | Low substitution rate and variant frequency | Negative Selection | Across species & human populations (~100 million years) | 98% |
| $F_{ST}$ afr-eas $F_{ST}$ afr-eur $F_{ST}$ eas-eur | Allele frequency differentiation between populations | Recent Positive Selection | Human populations (~ 75,000–50,000 years) | 99% |
| XP-EHH afr-eas XP-EHH afr-eur XP-EHH eas-eur | Cross-population extended haplotype homozygosity | Recent Positive Selection | Human populations (>10,000 years) | 83–86% |

performed in UK Biobank individuals for standing height (Table 2): Berg-2019 [30], Neale-2017 [38], GIANT-2018 [39], and Loh-2018 [40]. The four studies were selected to represent different methodological approaches. They were conducted in either unrelated white British individuals (Berg-2019, Neale-2017) or a more broadly defined population of European ancestry (GIANT-2018, Loh-2018). The Berg-2019 dataset is not corrected for population stratification, since they were evaluating its effects. The Neale-2017 and GIANT-2018 studies used ten genetic principal components while the Loh-2018 study used a linear mixed model (BOLT-LMM [41]) shown to be robust against population stratification. The GIANT-2018 meta-analysis had the largest sample size with 700K individuals whereas the other three had sample sizes of 335-460K individuals. The number of independent regions based on our LD-pruning approach increased with sample size except for the linear mixed model from Loh-2018 (n = 6,903), which was the highest (Table 1). The Benjamini-Hochberg p-value correction (p.adj) procedure was performed to control the false discovery rate (FDR) at 5% across 11 evolutionary measures for each GWAS.

Regions associated with height were enriched for signatures of sequence constraint (e.g. LINSIGHT, PhyloP, PhastCons) and differentiation between human populations ($F_{ST}$) in each of the four GWASs (FDR < 0.05, Fig 2A). Overall, nine out of the 11 evolutionary measures

**Table 2. GWASs on standing height used to evaluate robustness of our approach.** We used four published GWASs performed in the UK Biobank on standing height to evaluate the robustness of our approach. The year in the name is when the GWAS was published. Any correction for population stratification ("Stratification Correction") and the specific GWAS population ("Population") is noted. Using the same criteria for LD-pruning (Methods), we identified independent trait-associated genomic regions ("Independent Genomic Regions"). The Loh-2018 (40) GWAS used a linear mixed model (BOLT-LMM) shown to be robust against population stratification (41).

| GWAS Name | Stratification Correction | Population | Sample Size | Independent Genomic Regions |
|---|---|---|---|---|
| Berg-2019 | uncorrected | Unrelated white British | 337K | 2,505 |
| Neale-2017 | 10 PCs | Unrelated white British | 337K | 3,598 |
| GIANT-2018 | 10 PCs | European ancestry | 700K | 5,230 |
| Loh-2018 | Mixed effects Model | European ancestry | 459K | 6,903 |

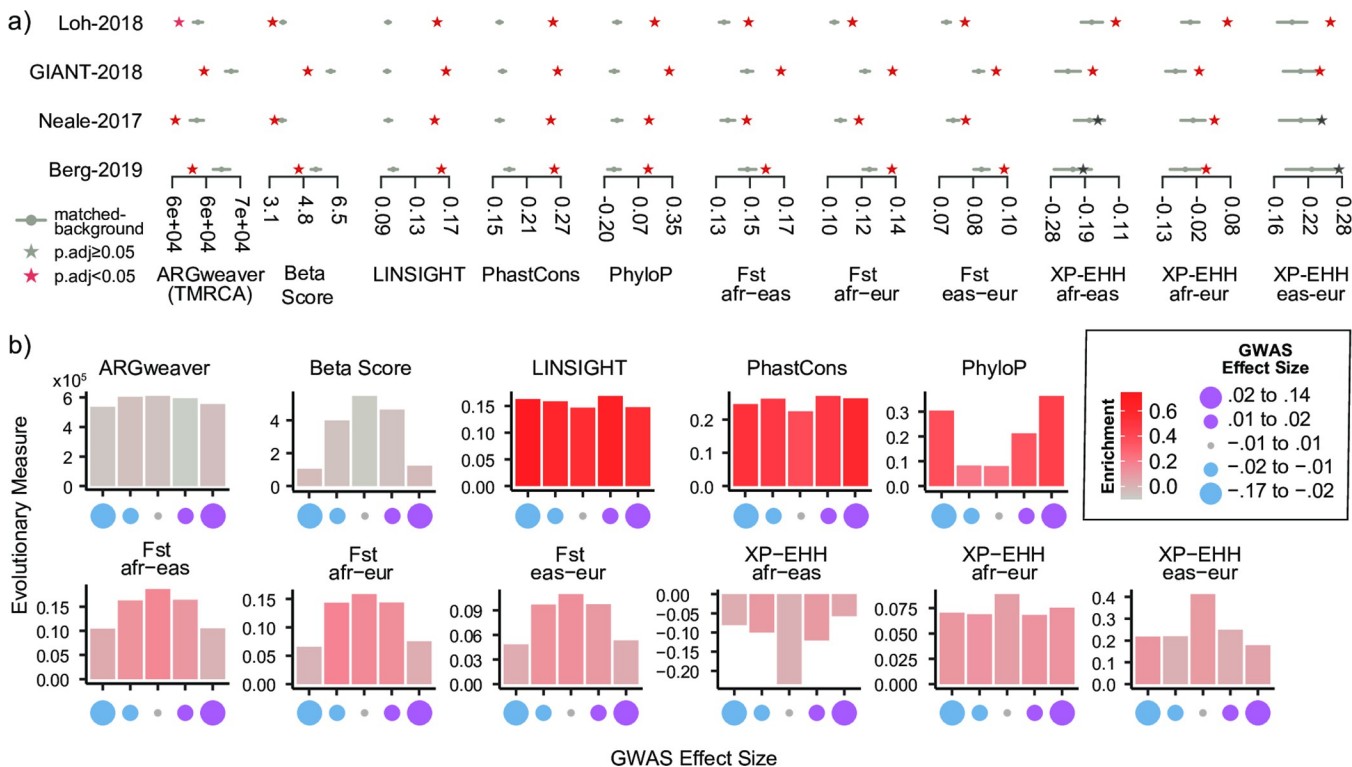

**Fig 2. The enrichment for evolutionary signatures is consistent across multiple GWASs of the same trait.** (a) For four separate GWASs of height (y-axis), we compared the mean trait-associated values (stars) for multiple evolutionary measures (x-axis) with their corresponding matched genomic background mean values (gray dot: mean value, gray bar: 5th, 95th percentile). We calculated an empirical p-value by comparing to the matched background (Methods) and adjusted for multiple testing (FDR-adjusted p-values < 0.05 are denoted as red stars, Methods). (b) For the Loh-2018 GWAS, we partitioned the trait-associated regions based on the association effect size (Beta Coefficient) of the lead SNP into five bins with equal numbers of trait-associated regions (x-axis). Each plot represents the mean value (y-axis) for a specific evolutionary measure. Bars are colored by their evolutionary enrichment values, which were calculated as described in Fig 1D. See Table 2 and Methods for details on the four GWASs analyzed.

had statistically significant deviations from the expected values (Fig 2A). These patterns, relative to the background distributions, were consistent across all GWASs and evolutionary measures. However, some measures (e.g. ARGWeaver, Beta Score, $F_{ST}$) showed greater variability for the mean observed trait value and background distributions than others (e.g. LINSIGHT, PhyloP, PhastCons). The two evolutionary measures (XP-EHH afr-eas and XP-EHH eur-eas) for which the statistical significance of the deviations from the background is not maintained across all four GWASs both measure population differentiation, and the two GWASs that do not show significant deviations (Neale-2017 and Berg-2019) both only include white British individuals.

We also randomly sampled trait-associated regions from the Loh-2018 GWAS without replacement to evaluate how evolutionary patterns varied based on the number of trait-associated regions. Across measures, we found that the background distribution and trait-associated value converged rapidly with an increasing number of trait-associated regions (S1 Fig).

These results also demonstrate the importance of matching the background distribution to the regions studied. For example, the observed Beta Scores for the Loh-2018 and GIANT-2018 regions are very different in magnitude (Fig 2A). Nonetheless, they are both similarly low compared to their appropriate background distributions. However, if the Beta Score values for GIANT-2018 had been compared to the Loh-2018 background distribution, we would have come to the opposite and incorrect conclusion that they were significantly higher than

expected. Overall, these results suggest that our approach is robust across GWASs and not substantially affected by their methodological differences.

## Some evolutionary signals vary across effect size

Based on evolutionary theory and recent observations [13], we expect stronger signatures of selection at regions with higher effect sizes. Thus, we stratified the trait-associated regions from the Loh-2018 GWAS into five bins with equal number of regions based on the GWAS effect size at each lead SNP. We observed several trends. Some evolutionary measures used to infer negative selection (LINSIGHT, PhastCons) had similar values and enrichment across bins (Fig 2B). For PhyloP, we observed higher values (suggesting conservation) for the largest effect size for trait-increasing variants (positive effect size) and for trait-decreasing (negative effect size) variants (Fig 2B). This is in line with the expectation that large effect variation occurs at conserved regions in the genome [42]. In contrast, measures often associated with local adaptation ($F_{ST}$), recent positive selection between human populations (XP-EHH), and balancing selection (Beta Score) had the highest values in bins with the smallest effect size (Fig 2B). Evolutionary enrichment was also strongest in bins with the smallest magnitude for $F_{ST}$ but generally similar across bins for XP-EHH (bar color, Fig 2B). When trait-associated regions were stratified by GWAS p-value (absolute effect size) instead, we generally saw similar trends with higher evolutionary measure values and enrichment for trait-associated regions with the smallest p-values (S2 Fig).

## A mosaic of diverse evolutionary forces on regions associated with complex traits

To generate an atlas of evolutionary signatures on complex-trait-associated regions, we analyzed the GWAS summary statistics of 972 traits (Methods). Summary statistics were downloaded from diverse sources including the Neale lab UK Biobank PheWAS (n = 202 traits) [38], the GWAS Catalog (n = 312) [43], GWAS Atlas (n = 297) [44], manual NCBI searches, and large consortia (Psychiatric Genomics Consortium, DIAGRAM, GIANT etc.). We applied our evolutionary enrichment computational framework to each GWAS. The resulting enrichments and trait-level statistics for eleven evolutionary measures can be downloaded from Fig-Share repository (https://doi.org/10.25452/figshare.plus.19733230) so researchers can explore traits of interest.

The number of trait-associated regions varied widely (mean: 183, median: 9, maximum: 5,678 regions). In our evolutionary atlas, 888 out of 972 traits had at least one trait-associated region meeting genome-wide significance (GWAS p-value < 5E-8). For traits with fewer than 50 associated regions, many (n = 432) lacked any statistically significant evolutionary enrichments (p-value<0.05 after multiple testing correction for the number of GWAS analyzed, Methods). Therefore, we focus here on describing evolutionary trends for traits (n = 290) with well-powered GWASs with 50 or more trait-associated regions. Out of the 290 well-powered GWASs, 231 were on quantitative traits (80%) and 59 were on discrete traits (20%).

For each evolutionary measure, we counted the number of GWASs with a significant deviation from the background (p-value < 0.05 after multiple testing correction for the total number of GWAS; Methods, S1 Table). Genomic signatures associated with negative selection were the most prevalent: 95% of GWASs had statistically significant enrichment for PhastCons (281/290), PhyloP (222/290), and LINSIGHT (278/290). We also commonly detected signals for the other modes of selection. More than half of the GWASs had significant enrichment for Fst (n = 152 to 194 traits), negative enrichment for balancing selection (Beta Score, n = 147 traits), and younger than expected allele ages (ARGweaver, n = 166 traits). Significant genomic

enrichment for signatures associated with cross-population positive selection (XP-EHH) were most prevalent for the African-European comparison (n = 138 traits) and less prevalent between Africans-East Asians (n = 37 traits), and Europeans-East Asians (n = 87 traits) comparisons. Though these differences may be driven in part by the bias towards European-ancestry individuals in genomic studies.

To illustrate the evolutionary patterns we observed across diverse traits, we plot the results for a subset of 47 GWASs carried out using the same BOLT-LMM mixed-effects model in the UK Biobank (Fig 3A) [40]. We refer to this analysis as the "BOLT-LMM set" (Methods). The BOLT-LMM set demonstrated the same general trends across evolutionary measures as we observed in the larger evolutionary atlas (Fig 3A and S1 Table). As examples of distinct evolutionary profiles, we highlight four traits: Age at Menarche, Sunburn Occasion (Sunburn), Hypothyroidism, and High Cholesterol (Fig 3B). Out of the four, age at menarche had the strongest enrichments for negative selection measures and negative enrichment for balancing selection and younger than expected allele ages. Sunburn's evolutionary profile was predominantly enriched for within human population genomic signals of differentiation (Fst, XP-EHH). Hypothyroidism had signatures of both negative selection and within human-population differentiation (XP-EHH). Similar to age at menarche, high cholesterol had strong signals of negative selection in addition to positive selection ($F_{ST}$, XP-EHH). Altogether, each trait is characterized by distinct evolutionary profiles.

## Skin and hair traits show signatures of local adaptation

Our analyses revealed strong enrichment for evolutionary measures associated with local adaptation for GWASs of hair and skin traits (Fig 3A). In the BOLT-LMM set, the GWASs for hair color traits were highly polygenic with over 1,000 trait-associated genomic regions. GWASs for skin-related traits (sunburn, tanning, skin color) had variable degrees of polygenicity (34 to 854 trait-associated regions), while the GWASs for the two balding traits had around 700 trait-associated regions. Except for the GWAS for the tanning trait, all others demonstrated strong positive enrichment for signatures of negative selection (LINSIGHT, PhastCons, Fig 3A). They also exhibited strong positive enrichment for $F_{ST}$ across the three 1000 genomes superpopulations. Hair/skin color and tanning trait-associated regions had enrichment of signatures of recent positive selection in the European superpopulation (negative XP-EHH afr-eur) compared to the African superpopulation. Meanwhile, the balding trait-associated regions had enrichment for signatures of recent positive selection in the African superpopulation compared to the European. Similarly, evidence of recent selection between African and East Asian superpopulations was observed for GWASs of dark hair and skin color. Recent selection between East Asian and European super populations was observed for GWASs of hair color, skin color, tanning and sunburn.

## Alzheimer's disease associated genomic regions lack enrichment for evolutionary signatures

The GWASs of nearly all traits in the BOLT-LMM set had enrichment for diverse genomic signatures of selection. In contrast, we observed that genomic regions associated with late-onset Alzheimer's disease exhibited no significant enrichment for any evolutionary measure (Fig 4). This result held across five published GWASs of late-onset Alzheimer's disease. The GWASs had between 19 to 132 trait-associated regions identified in European-ancestry populations: Bellenguez [45], Marioni ([46], Kunkle [47], GRACE [48], IGAP [49]. Across the 11 evolutionary measures we tested, all GWASs had trait-associated evolutionary values that overlap the expected range from their matched backgrounds (p>0.05, Fig 4). Consequently, we did not

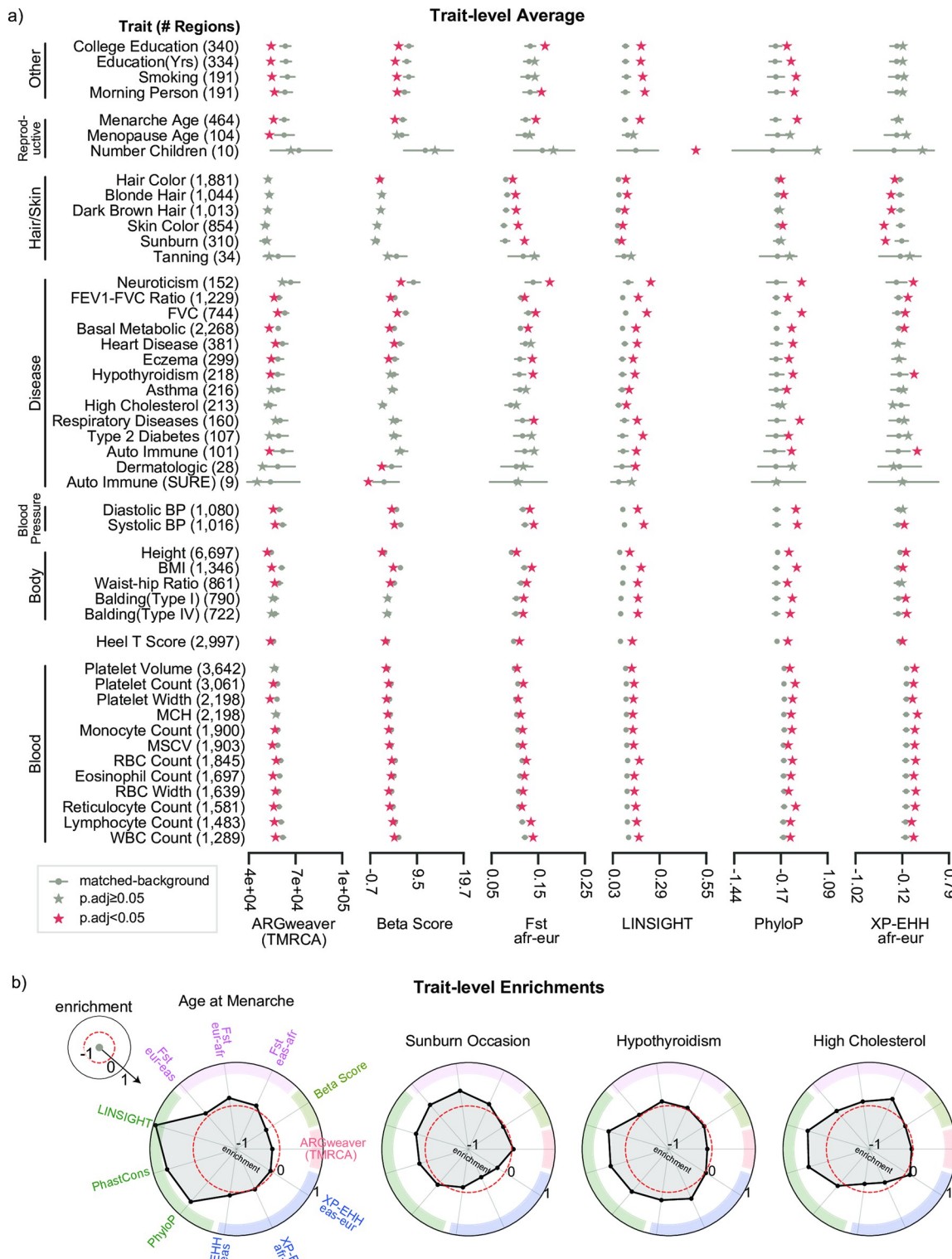

**Fig 3. Mosaic evolutionary architecture across 47 well-powered GWASs of human complex traits.** From our evolutionary atlas of 972 GWASs, we plot a subset of 47 GWASs (BOLT-LMM set) performed using the same approach and from the same cohort (Methods). (a) For each evolutionary measure (columns) and a given trait (row), we calculated the trait-averaged value (x-axis, stars) and compared it with the matched genomic background distribution (gray dots: mean values, gray bars: 5th, 95th percentiles). Traits are manually grouped based on type and similarity. The number of trait-associated regions is provided in parentheses. Red stars

(FDR<0.05) represent statistically significant deviation after multiple testing correction (Methods). Results are shown for six evolutionary measures; see S3 Fig for all 11 evolutionary measures. (b) We calculated enrichment as described in Fig 1D and highlight four traits with distinct evolutionary profiles. Spokes represent different evolutionary measures (colored by type of associated force) and concentric rings represent levels of evolutionary enrichment. Red dashed circles represent the expected values (i.e., no enrichment).

detect any genomic signatures of enrichment across the evolutionary measures we tested. Thus, we hypothesize that genomic regions associated with some late-onset traits may be less likely to have strong signatures of selection. We mined our dataset for comparable late-onset traits, especially those with a neurological component, but we did not find any that met our quality threshold of 50 or more associated genomic regions. GWASs for other relatively late onset diseases like coronary artery disease showed positive enrichment for many signatures as observed for most traits (Supplementary Figshare Repository https://doi.org/10.25452/ figshare.plus.19733230). Analysis of additional high powered GWASs is needed to determine the role of age-of-onset in the enrichment of the studied evolutionary metrics.

## Discussion

Natural selection has influenced patterns of variation in genomic regions associated with many human complex traits. However, the role of different modes of selection, neutral processes, and the extent of their influence on genomic regions associated with complex human traits remain challenging to study. Here, we couple the availability of summary statistics from 972 GWASs with 11 evolutionary statistics to identify enrichment for evidence of different evolutionary forces on genomic regions that contribute to variation in the human phenome. Our empirical approach quantifies enrichment compared to background genomic regions matched to those identified for each trait. The analysis pipeline can flexibly incorporate any evolutionary measure with genome-wide SNP level annotation and quantify a trait-level summary and enrichment. We make our evolutionary atlas and efficient open-source software available for the research community (https://github.com/aa-publications/gsel_vec).

We observe several consistent trends across regions associated with diverse complex traits. Evolution measures associated with negative selection, both within and between species, are enriched among variants associated with nearly all complex traits. This indicates that, as expected, trait-associated variation is enriched in functional regions with significant evolutionary constraint. We also consistently observe significantly younger ages for trait-associated alleles, which suggests that recent variants make a substantial contribution to the common-variant mediated variation in most complex traits. We also observe enrichment for signatures of differentiation/positive selection between populations for a substantial fraction of traits, most notably those involved in hair, skin, blood measurements, and the immune system. This

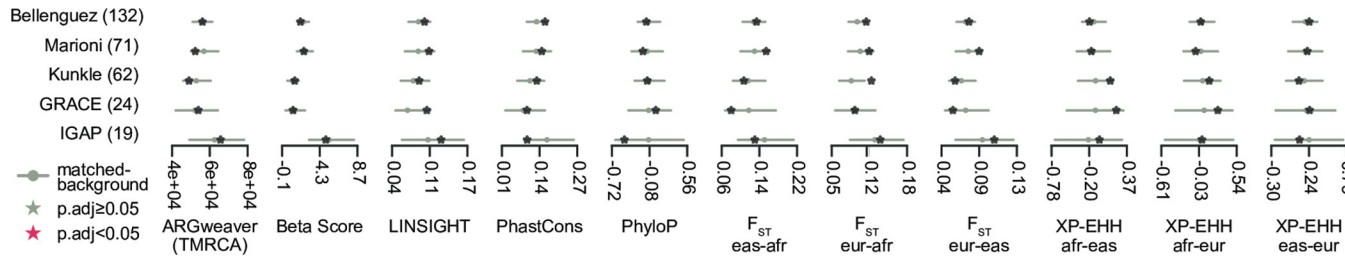

**Fig 4. Loci for late-onset Alzheimer's disease lack enrichment for evolutionary forces.** Across five GWASs conducted on Alzheimer's Disease (y-axis), we plot the trait-averaged value (red or black stars) across evolutionary measures (x-axis) compared to their matched genomic background values (gray bars, 5th, 95th percentiles. We did not find significant enrichment for any evolutionary measures (FDR<0.05 with multiple testing correction, Methods). This pattern held across all five GWASs considered. This suggests that genomic regions contributing to the development of Alzheimer's Disease are not enriched for specific evolutionary forces.

is consistent with recent population-specific adaptation driven by these traits with particular relevance to survival in new environments [11]. We note, however, that multiple evolutionary forces can influence any one metric, making direct connections between a summary statistic and the evolutionary process difficult. For example, $F_{ST}$ is generally used as a metric to identify variants with greater than expected population differentiation. This metric is also influenced by background selection [50], mathematical constraints [51], and balancing selection [52]. Nonetheless, regions associated with most traits show strong enrichment for multiple evolutionary patterns, suggesting that a mosaic of selective pressures commonly shaped variants associated with complex traits.

Our approach generalizes the common strategy of analyzing evolutionary patterns on individual loci of interest to comprehensively characterize all regions associated with a trait. This region-focused approach has several advantages. Previous empirical work [14,35] has shown the promise of quantification of region-level pressures to understand evolutionary forces on a handful of traits and interpret associated loci. Calculating a standardized enrichment for each trait and measure from an appropriate background enables us to compare across different evolutionary measures and, consequently, generate evolutionary profiles across GWASs of different traits. Our findings are consistent with several recent genome-wide analyses that use different approaches and identify widespread global differentiation [35], negative selection [13], and polygenic adaptation [53] on complex traits.

Differences in the average polygenic risk score between populations and the correlation between polygenic risk scores and geographic clines [27–29] or time [53] have been used to argue for polygenic adaptation on traits such as height. However, such approaches can yield false signatures of adaptation due to inflated differences arising from population stratification in the GWASs [30,31,54]. Our approach is distinct from and complementary to recent methods for detecting polygenic selection from GWAS in several key aspects. First, it separates the identification of genomic signatures of different evolutionary forces from the trait(s) that drove the selection. While both are challenging problems, identifying the specific traits driving selection is not necessary to infer that selection occurred in genomic regions associated with these traits. Rigorous detection of polygenic adaptation would require detailed phenotypic and environmental measurements over time and/or across different populations. The difficulties accounting for stratification in previous studies of height illustrate these challenges. Such an approach is not currently possible at scale since both modern and ancient phenotype data are very sparse for most traits and many of these pressures happened deep in our evolutionary history. Thus, our atlas provides a complementary high-level overview of the currently detectable evolutionary signatures on genomic regions that underlie complex traits. We anticipate that this can help generate hypotheses about which traits may have experienced different selective pressures.

A second major difference is that we do not directly consider effect size or direction inferred from GWAS therefore reducing the potential effect of inflated or unstable estimates between populations. However, we acknowledge that trait-associated regions near the significance threshold could still be enriched for false positives due to population stratification. Nonetheless, our framework enables us to evaluate the relationship between effect size and evolutionary signatures of selection (Fig 2). We observe for height that the most extreme scores and strongest enrichment for evolutionary measures associated with differences between human populations ($F_{ST}$, XP-EHH) occur at lower effect sizes.

Third, by summarizing the distribution of evolutionary measures at the local region and then genome-wide level, we obtain a richer characterization rather than considering a single tag SNP, which may be subject to substantial variation and not truly causal. Moreover, this allows us to build an appropriate background distribution. This is especially important, since

the strength of selection is not uniform but often varies based on functional annotations across the genome [13,55]. We are also able to corroborate observations by incorporating multiple evolutionary measures capturing similar evolutionary forces (e.g., PhyloP, PhastCons, LIN-SIGHT). Finally, our framework flexibly considers signatures of many different evolutionary forces, not just adaptation. We are also able to compare the enrichment for signatures of selection across traits and effect sizes.

Our approach also has some limitations. First, as noted above, if the goal is to find traits under selection, then identification of selection acting on genomic regions associated with a trait does not necessarily imply that selection acted on the trait itself. Linking genomic signatures of selection to traits is complicated by pleiotropy, especially antagonistic pleiotropy, e.g., regions associated with heart disease and lifetime reproductive success exhibit antagonistic effects [56]. Furthermore, the omnigenic model suggests that pleiotropy is extremely pervasive across human traits [57]; thus, attributing the contributions of selection on different genomic regions to individual traits is likely to be a considerable challenge. Second, rare variants contribute to variation in many complex traits [58], and our use of GWAS data limits our analyses to relatively common variants. Nonetheless, our approach can be used to analyze known rare variants, and increasing GWAS sample sizes are enabling the detection of effects for increasingly rare variants. Third, lower frequency alleles tend to have larger effects in GWAS [59], even in the absence of any selection-driven relationship. These relationships between allele frequency and LD with power to detect associations and evolutionary signatures was a major motivation for our approach of developing empirical background distributions after matching on these relevant attributes. However, it is difficult to verify that these effects have been fully accounted for, especially with regard to variants that are challenging to detect in GWAS. These considerations are particularly important in the context of the few analyses that consider allele frequency (Fig 2B). Fourth, many of the traits we examined are polygenic. Whether the evolutionary trends we observe can be generalized for oligogenic traits warrants further study.

Finally, the limited availability of GWAS data from non-European populations [60] required us to focus on trait-associated regions identified in European populations. This lack of diverse GWAS data limits our understanding of the extent of shared genetic architecture across human populations for complex traits. It is likely that trait-associated regions identified in European populations share similar biological function and trait-relevance in non-European populations. However, effect sizes are often not generalizable across populations [61]. Thus, we elected to not directly consider effect size in most of our analyses, but the results of between population statistics in Fig 2B are sensitive to this assumption.

The flexibility of our approach enables several future directions. As new evolutionary measures are developed, they can easily be integrated into our framework. Our approach could also be integrated with model-based evolutionary simulations to better understand the effects of different evolutionary pressures and their combinations on the distributions of the statistics. Evolutionary enrichment at the trait level can be used to better understand pleiotropy and whether the enrichment varies across functional regions of the human genome for a given trait. As more diverse GWASs conducted in non-Europeans become available, our framework can be used to compare genomic signatures of selection across human populations. This will enable additional tests for evidence of polygenic adaptation, such as heterogeneity among loci and non-parallelism between replicated populations [62]. Additionally, our framework is not limited to the human species; the same approach can be applied to GWAS conducted in any species such as mice [63], non-human primates [64], or fungi [65]. In summary, our quantification of genomic signatures of selection on trait-associated regions advances our understanding of the genetic architecture of complex traits and illuminates the diverse forces that have shaped functional regions of the human genome.

## Methods

### Detecting genomic signatures of evolutionary forces from summary statistics

Our empirical framework to detect evolutionary signatures relies on building a matched background to compare trait-associated regions. For a given trait, we identify independent trait-associated regions by pruning using LD (r2≥0.9), genomic distance ≤ 500 kbases, and GWAS p-value < 5E-8 (Fig 1A). This is obtained by running the—clump flag in PLINKv2 with the following parameters:—clump-kb 500,—clump -r2 0.9,—clump-p1 5E-8,—clump-p2 5E-8. We refer to the independent regions identified by LD clumping as trait-associated regions and variants with GWAS p-value < 5E-8 within the clumped regions as trait-associated variants. All genomic coordinates are GRChg37.

For each trait-associated region, we match using an approach motivated by SNPSNAP [66] and described previously [14]. Briefly, for each lead variant (variant with lowest p-value) in a trait-associated region, we randomly select 5,000 control variants matched on the following features: allele frequency (+/-5%), LD (r2>0.9, +/-10% LD buddies, gene density (+/− 50%) and distance to nearest gene (+/−50%) (Fig 1B). We implemented the matching as a python script. Matched variants were drawn from 1000 Genomes subset of the European superpopulation.

To extend the LD matching for each trait-associated region, if more variants are present in LD with a matched variant, we randomly select variants in LD. For example, in a given trait-associated region, if there are four trait-associated variants in LD (r2>0.9) with the lead variant, then for each of the 5,000 matched control variants, we select 4 variants in LD (r2>0.9) with each. This pairs each of the 5,000 matched control variants with the same number of LD variants as the trait-associated regions. Matching is attempted five times starting with the strictest threshold for each metric (e.g. MAF +/- 1%) and increasing incrementally to the most liberal threshold (e.g. MAF +/- 5%). In the rare case that there are still no matched regions, the lead GWAS variant is removed from subsequent analysis. Additional detail on the method can be found in the original publication [14].

Next, for every evolutionary measure, we calculated a trait-level average using two steps. First, we calculate for each region (matched or trait-associated) a 'region-average' defined as the greatest absolute value across all trait-associated variants. For the second step, we calculate the trait-level average across all the region-averages for the trait-associated regions and each of the 5,000 matched sets, where each set includes a matched region for each trait-associated region (Fig 1C). The 5,000 averaged evolutionary values make up the background distribution that we use to compare the trait-average evolutionary measure value (Fig 1D). We derive unadjusted p-values by quantifying the number of averaged matched evolutionary values as or more extreme than the trait-average out of the 5,000. We adjust this p-value for multiple testing in each analysis. Additionally, using this background distribution, we define evolutionary enrichment as the difference between the trait-level mean and the mean of the background distribution divided by the genome-wide standard deviation of the evolutionary measure (Fig 1E). This standardization allows us to compare the relative enrichment across different evolutionary measures. In summary, this approach starts with GWAS summary statistics and quantifies a trait-level average and enrichment for a given evolutionary measure.

### Source of evolutionary measures

In this study, we downloaded or calculated eleven evolutionary measures (Table 1) for all trait-associated and matched control variants as described in our previous study [14]. Briefly,

VCFTools (v0.1.14) [67] was used to calculate pairwise $F_{ST}$, the R package rehh (v2.0) was used to calculate XP-EHH using phase 3 1KG data. BetaScan software [19] was used to calculate Beta Score. PhyloP [68], PhastCons 100-way [69], LINSIGHT [70], and Allele Age [21,71] were downloaded from their publications or the UCSC Table Browser [72].

### Evaluating robustness of evolutionary signatures using height GWAS summary statistics

GWAS summary statistics for standing height were downloaded from four different studies (Table 2). The Berg-2019 analysis performed a linear regression with age, sex, and sequencing array as covariates on unrelated British ancestry individuals in the UK Biobank [30]. We downloaded the summary statistics labeled "UKBB_noPCs" from datadryad.org/stash/dataset/doi:10.5061/dryad.mg1rr36. The Neale-2017 analysis also performed a linear regression with the first genetic 10 principal components and sex as covariates on unrelated white British individuals [38] [cite: http://www.nealelab.is/uk-biobank/]. Summary statistics were obtained by downloading the file 50_raw.gwas.imputed_v3.both_sexes.tsv from the "GWAS round 2" repository hosted at nealelab.is/uk-biobank. The GIANT-2018 summary statistics were obtained from a meta-analysis of previous height GWAS on European ancestry combined with the UK Biobank cohort that included age, sex, recruitment center, genotyping batches and 10 genetic principal components [39]. The summary statistics were downloaded from: https://portals.broadinstitute.org/collaboration/giant/images/6/63/Meta-analysis_Wood_et_al%2BUKBiobank_2018.txt.gz/. The Loh-2018 analysis used a linear mixed model on individuals of European ancestry from the UKBiobank [40]. The height summary statistics were downloaded from https://alkesgroup.broadinstitute.org/UKBB/ (file name: body_HEIGHTz.sumstats.gz).

On all four summary statistics, we applied our approach to detect genomic signatures of evolutionary forces. We calculated a trait-associated region average and the distribution of the background set and the evolutionary enrichment as described earlier (Fig 1). For each summary statistic, we corrected for multiple testing across the 11 evolutionary measures using the Benjamini-Hochberg FDR control approach.

To test for effects of trait-associated p-value obtained from the summary statistics, we created quintiles with an equal number of trait-associated regions based on the GWAS summary statistics p-value at the lead SNP. We then applied our evolutionary analysis on each quintile. We repeated the same steps to test for the effect size from the GWAS summary statistics but instead created quintiles based on the beta coefficient. We also tested whether the enrichment calculation was sensitive to matching parameters for gene density and distance to nearest gene. To do this, we repeated our enrichment calculation on each quintile based on effect size for either gene density +/- 25% and +/- 10% or gene distance +/- 25% and +/- 10% (S4 Fig). When the matching thresholds are varied, the enrichments in evolutionary metrics do not vary substantially from the original analysis. This suggests our enrichment calculation is not sensitive to matching parameters.

To test how the number of trait-associated regions affected our evolutionary analyses, we randomly sampled with replacement the number of trait-associated regions to create under-sampled sets. Then for each set, we ran our evolutionary pipeline to calculate a trait-level average (S1 Fig).

### GWAS datasets to generate evolutionary atlas

We used multiple sources to identify GWASs that were conducted in individuals of European ancestry and had complete publicly available summary statistics for all analyzed regions

reported in human genome version hg19. GWASs were curated from repositories such as the Neale Lab analysis of the UK Biobank data [38], the GWAS catalog [43], the GWAS Atlas [73], major GWAS consortia such as the Psychiatric Genomics Consortium and DIAGRAM, and manual NCBI searches for specific traits. Our inclusion criteria were that the study was conducted in a European population and the complete summary statistics were freely available for download and reported in the hg19 assembly. We excluded GWASs that only reported the top hits. Details for each of the GWAS summary statistics in our evolutionary atlas are provided in S1 File, which includes the PMID of the study and the web link to download the raw summary statistics.

For each summary statistic, we applied our approach to detect genomic signatures of evolutionary forces as described earlier. GWASs without any significant independent regions (based on p-value and LD as described above) were not further analyzed. For all GWAS with at least one associated region we retained the summary statistics for every individual trait-associated genomic region and the trait-level enrichment across the entire GWAS. To correct for multiple testing, empirical p-values across all traits for a given evolutionary measure were adjusted using the Benjamini-Hochberg FDR control approach.

This data is available on FigShare reports empirical p-value only and should be adjusted accordingly for future analyses (https://doi.org/10.25452/figshare.plus.19733230).

### BOLT-LMM GWASs subset analysis

We further analyzed a subset of 47 traits, which we refer to as the "BOLT-LMM set", whose summary statistics were generated using a mixed modeling approach [40]. All summary statistics were downloaded from https://alkesgroup.broadinstitute.org/UKBB/. We ran our evolutionary analyses to calculate trait-level averages and the background distribution (Fig 3A). Empirical p-values were corrected for multiple testing across traits and evolutionary measures using the Benjamini-Hochberg FDR control method. Next we calculated the evolutionary enrichment for each trait and evolutionary measure.

### Late-onset Alzheimer's disease analyses

We performed our evolutionary analysis on five GWAS of the late onset Alzheimer's trait. The GWAS analyzed were collected from the following sources: Bellenguez et al.[45], Marioni et al. [46], Kunkle et al. [47], GRACE [48], IGAP [49]. The most recent GWAS (Bellenguez et al.) was reported in hg38 and converted to hg19 using the biomaRt (v4.2) package in R [74] using the archived Ensembl 75: Feb 2014 (GRCh37.p13). Empirical p-values were corrected for multiple testing across all five GWAS and 11 evolutionary measures using the Benjamini-Hochberg FDR control approach.

## Supporting information

**S1 Table. Count of traits with signals of evolutionary forces.** Number of traits ("# Traits") in the full Evolutionary Atlas (top) and BOLT-LMM subset (bottom) with statistically significant enrichment for evolutionary measures (rows). Note, only traits with 50 or more associated regions are analyzed within the Evolutionary Atlas. The proportion out of all traits analyzed ("Proportion of All Traits (%)") are shown for the Evolutionary Atlas (n = 290 traits) and BOLT-LMM set (n = 47 traits). Depletion refers to negative enrichment.
(DOCX)

**S1 Fig. Evolutionary signatures converge rapidly with increasing number of trait-associated genomic regions.** Using the Loh et. al. GWAS, we randomly undersampled the number

of trait-associated regions without replacement (x-axis) and measured the mean evolutionary measure at trait-associated regions (blue line) and the matched background (mean: black line, gray shading between 5th and 95th percentiles). The observed evolutionary measures for trait-associated regions and their relative values compared to the matched background regions are consistent across different numbers of associated loci considered.
(TIF)

**S2 Fig. Strongest genomic evolutionary signatures occur in most significant trait-associated regions.** Using the Loh-2018 (Fig 2) GWAS, we partitioned trait-associated regions into five bins with equal number of regions based on GWAS p-value of the lead SNP in each region. Each plot represents the mean trait value (y-axis) for an evolutionary measure and each bar is colored by the evolutionary enrichment which is calculated as described in Fig 1D.
(TIF)

**S3 Fig. Mosaic evolutionary architecture across 47-well-powered GWASs across 11 evolutionary measures.** On a subset of 47 GWASs (y-axis, BOLT-LMM set), the trait-level average (red star or gray 'x') for 11 evolutionary measures (x-axis) compared to its matched background distribution (gray dots: mean values, gray bars: 5th, 95th percentiles) are displayed. The number of trait-associated regions is provided in parentheses. Red stars (p.adj<0.05) represent statistically significant deviation after multiple testing correction (Methods). This figure extends Fig 3A by including all 11 evolutionary measures considered in this study.
(TIF)

**S4 Fig. Enrichment for evolutionary measures is not sensitive to different matching parameters for gene distance and density.** For each evolutionary measure (one plot per measure), we repeated the analysis in Fig 2B and calculated the enrichment (y-axis) across trait-associated regions partitioned by association effect size (x-axis, ordered from negative to positive effect size) for the original Fig 2B analysis (red X) and four other conditions. We repeated the analysis by changing either the gene distance matching threshold to be either +/- 25% (dark blue) or +/-10% (light blue) or the gene density matching threshold to be either +/- 25% (dark green) or +/-10% (light green) while keeping the all other parameters the same. The patterns are similar for nearly all settings.
(TIF)

**S1 File. File_S1.xlsx** This excel file contains PMID or web link and the source for each GWAS summary statistics analyzed in this study.
(XLSX)

## Acknowledgments

This work was conducted in part using the resources of the Advanced Computing Center for Research and Education at Vanderbilt University.

## Author Contributions

**Conceptualization:** Abin Abraham, Abigail L. LaBella, John A. Capra, Antonis Rokas.

**Data curation:** Abin Abraham, Abigail L. LaBella.

**Formal analysis:** Abin Abraham, Abigail L. LaBella.

**Funding acquisition:** Abin Abraham, John A. Capra, Antonis Rokas.

**Investigation:** Abin Abraham, Abigail L. LaBella, John A. Capra, Antonis Rokas.

**Methodology:** Abin Abraham, Abigail L. LaBella.

**Project administration:** John A. Capra, Antonis Rokas.

**Resources:** John A. Capra, Antonis Rokas.

**Software:** Abin Abraham, Abigail L. LaBella.

**Supervision:** John A. Capra, Antonis Rokas.

**Validation:** Abin Abraham, Abigail L. LaBella.

**Visualization:** Abin Abraham, Abigail L. LaBella.

**Writing – original draft:** Abin Abraham, Abigail L. LaBella.

**Writing – review & editing:** John A. Capra, Antonis Rokas.

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
