## [Decision Letter · Decision Letter 0]

29 Jul 2022

Dear Dr Rokas,

Thank you very much for submitting your Research Article entitled 'Mosaic patterns of selection in genomic regions associated with diverse human traits' to PLOS Genetics.

The manuscript was fully evaluated at the editorial level and by independent peer reviewers. The reviewers appreciated the attention to an important topic and they were all positive about work. They also identified some concerns, and have commented on areas that would improve the presentation and quality of the work. We therefore ask you to modify the manuscript according to the review recommendations. Your revisions should address the specific points made by each reviewer.

[LINK]

Yours sincerely,

Justin C. Fay

Academic Editor

PLOS Genetics

Bret Payseur

Section Editor

PLOS Genetics

Reviewer's Responses to Questions

**Comments to the Authors:**

Reviewer #1: With their work the authors aim at providing a comprehensive overview of evolutionary signatures from a number of traits targeted by GWAS available from the literature.

The effort is appreciable as an evolutionary perspective on GWAS signals can indeed provide orthogonal knowledge when interpreting the association signals.

The work is clearly written and definitely deserves to be published, however there are three assumptions that should be clarified and/or addressed for the work to be fully understandable to a non specialized reader:

1) The vast majority of GWAS hits were retrieved using West Eurasian populations. Using them to pinpoint genetic regions linked to a given loci in non European populations (e.g. East Asians and Africans) introduces the assumption that the loci "tagged" by the West Eurasian GWAS are involved in the biological processes related to a given trait in all human populations. In other words, that the biology underlying a given trait is the same across all pops. This is an assumption, and as such it should be spelled out. As an example, other papers that adopted a similar approach ( https://www.sciencedirect.com/science/article/abs/pii/S0960982222001087 ) only investigated European populations, for which this assumption can surely be granted.

2) Even taking for granted the above assumption for all human pops, surely the effect sizes (discovered in West Eurasians) were shown to be not fully transferable across pops (see Martin et al. 2017 and other relevant papers). It follows that Figure 2b (and related findings) for non European populations ( Afr-Eas, but also Eur-Eas and Afr-Eur pairs, for examples) may leverage on a weak assumption (e.g. that binning by effect sizes would yield the same binning in all the tested pops);

3) The result that highly polygenic traits such as height are enriched for signatures of negative selection make me wonder how well "gene content" was controlled for when creating the matched-sets. I see you have a +/- 500% parameters in your recipe: perhaps you could try whether the negative selection enrichment holds when using a more conservative 200% (or some other type of matching: https://www.cell.com/ajhg/fulltext/S0002-9297(13)00127-4)? Or would this leave you with little or none matched SNPs? I agree that finding a matched set for a highly polygenic trait is hard; on the other hand the report of negative selection should be weighted carefully (or toned down if you cannot be sure whether you properly controlled for gene content) to avoid giving the impression that these traits are indeed under negative selection.

Minor points:

1) Line 61: negative selection decreases the apparent substitution rate, but I am not quite sure positive selection increases it.

2) Figure 2b: I might have missed it, but could you comment more on the U shaped pattern of PhyloP? Is the U shape expected, or is it quite promising even though there is no difference in enrichment.

Reviewer #2: In this manuscript, Abraham et al. have generated wide-ranging and valuable resource: no less than 11 measures of natural selection calculated for over 900 GWAS. Subsequent analyses focus on general patterns that are observed for 290 high power GWAS. Although there is some risk of this study feeling like everything but the kitchen sink was included - the authors managed to pull it off and write a coherent paper. This manuscript is well written, and the multi-panel figures manage to convey a large amount of information in a concise way. Reasonable LD pruning and p-value thresholds were used for inclusion of trait-associated SNPs (the former is particularly important given that GWAS hits need not be causal SNPs). The authors also appear to have correctly implemented selection scans for each of the 11 evolutionary measures. This manuscript fits well within the scope of PLoS Genetics and is likely to be of interest to a broad readership. However, some additional analyses may be needed before this manuscript would be acceptable for publication.

One key issue is that the authors adopted outlier approaches. While there is nothing inherently wrong with this, I wonder about how many of the signatures of selection could be false positives (especially since a modest adjusted p-value cutoff of 0.05 was used). How to get around this? One option would be to use a more stringent adjusted p-value cutoff. A second option would be to use SLiM computer simulations that incorporate reasonable demographic models, and testing how each of the 11 different evolutionary measures perform. This latter option would take a lot more work to implement - and I would definitely be satisfied if the authors simply chose the first option.

A second key issue involves the inclusion (or lack thereof) of effect sizes. This manuscript would be greatly improved if the authors would test how well their z-scores (what the authors call "evolutionary enrichment values") are robust to including weightings of trait-specific effect sizes - i.e., relaxing point (1) on line 104. Although the authors correct that stratification and ascertainment bias is a possible confounding factor, it is important to keep in mind that some traits have genetic architectures that are driven by a small number of loci of large effect, and any tests of selection should take this into account.

Additional points:

The authors should be a bit more explicit in the sources of GWAS results, including criteria for inclusion of a particular GWAS. How many of the 290 well-powered GWAS for quantitative traits and how many are for discrete or binary traits?

Line 81: The passage that includes "the genomic background does not provide an appropriate null when interpreting overlaps between trait associations and signatures of selection" is a little unclear especially since line 105 includes the phrase "builds a null distribution from allele frequency and LD-matched SNPs." The authors are advised revise for clarity.

"Evolutionary enrichment" seems a little bit of a misnomer. tThat said, the approach used by the authors yields a reasonable summary statistics. I wonder if "Evolutionary z-score" might be a better way of writing things.

I like that their approach was tested on height. However this is an exceptionally polygenic trait, (and therefore less likely to be affected by the decision to not directly incorporate effect sizes - as per line 104).

Table 1: It would be advisable to list the actual time scale (in terms of ky or My for each of these evolutionary measures. Also, "Evolutionary origin" seems like an odd term here. Maybe change it to something like TMRCA and keep ARGweaver in the first column?

Figure 2b: It seems overly redundant to partition things based on negative and positive effect size (as evidenced by the symmetry of each of the subplots). It makes much more sense to partition using the absolute value of effect size (perhaps at finer resolution to retain five bins per evolutionary measure). Alternatively, the authors could partition trait-associated SNPs by p-value or MAF.

Figure 3b: enrichment scores of -1 or +1 are still pretty modest. Why doesn't the enrichment score need to be > |2| for things to be significant?

Line 353: Great hypothesis. Why not explicitly test whether genomics regions that are associated with late-onset traits are less likely to have strong signatures of selecting with the data that is in hand? Adding a small analysis along these lines would improve this paper.

Reviewer #3: Abraham & LaBella et al develop a new framework for analyzing the signals of selection underlying complex trait-associated variation. They apply this method to human GWAS summary statistics and genomic data and describe several findings pertaining to differences between associated regions and background control regions for several statistics such as Fst, XP-EHH, and PhyloP.

I found the manuscript quite interesting, and it has several strengths, including a novel framework for sampling genomic background regions that could prove useful in future research, some interesting observations about potential evolutionary mechanisms that might underly the enrichments/depletions they detect, and an appreciated review of some limitations in the discussion section. I could see myself using their approach, especially paired with model-based simulations to explore the effects of evolutionary processes on the joint distribution of these statistics.

I do have some comments about the interpretation of the observations in this manuscript. In general, the authors have described this method as a way to characterize evolutionary processes underlying trait variation, and they have categorized the various statistics they use (e.g., Fst) by specific evolutionary processes (e.g., positive selection). However, most evolutionary processes (different modes of selection, demographic processes, recombination, mutation rate changes, etc) will have some impact on these statistics, and hence it is difficult for me to see how differences in the distributions between background and foreground sets could be ascribed directly to specific processes.

Differences between foreground and background sets can be described as consistent with the action of various selection modes (e.g., differences in Fst are consistent with positive selection), but other evolutionary processes usually cannot be ruled out based on this type of analysis. To me this is a very important distinction, related to several ongoing debates in the field about the relative impact of various evolutionary processes on genomic variation. I provide a few specific examples below, followed by some minor comments. I think this main concern could be addressed primarily by changing the wording throughout the manuscript. The method could be described as an approach for rigorous assessment of enrichment/depletion of evolutionary patterns, possibly driven by various evolutionary processes.

Main comments:

As I’m sure the authors are aware, multiple different evolutionary processes could affect the summary statistics that they have included in their analysis. I’m going to briefly itemize some studies that specifically address how these statistics are affected by processes that could differ between background and foreground sets.

1. Haplotype scans may be affected by demographic processes and multiple modes of selection (Tadashi et al 2006 Genome Research). While the whole genome experiences the same demography in a panmictic population, the same is not necessarily true in a real GWAS sample with any mixed ancestry or admixed individuals. (Variation in background selection will also cause the effects of demography to be experienced differently by different parts of the genome, as in Torres 2018, noted below). XP-EHH differences might be affected by such processes and not only due to positive selection.

2. Fst is affected by both background selection (Torres et al 2018 Plos Genetics) and mathematical constraints on the values it can take (Jakobsson et al 2013 Genetics). Fst can be elevated under positive selection, but causal SNPs may also be excessively differentiated under other selection modes (e.g., stabilizing selection w/o any change in the mean, Yair & Coop, Phil Trans B 2022).

3. Balancing selection is likely to generate high diversity and short haplotypes, and hence is very likely to affect several of the statistics used here. Fst has been used to detect balancing selection amongst other modes of selection (Foll & Gagiotti 2008 Genetics; see also Ecxcoffier 2009 Heredity) under the principle that allele frequencies should be more similar than expected. However, if the environment changes but balancing selection is maintained, the equilibrium frequency of the balanced allele will change, resulting in unknown (and unpredictable) effects on Fst.

These are just some examples — in general I don’t think it’s possible to directly ascribe an evolutionary process to differences in the distributions of these statistics. The authors could revise the manuscript by clarifying how their findings are consistent with various modes of selection, rather than fully explained by these evolutionary processes. They could also include a short section describing the ways that multiple evolutionary processes could underly some of the differences in these statistics in the discussion.

In addition, I think that the finding that effect sizes are correlated with the values of some statistics could be further justified (Fig 2b). As I’m sure the authors are well aware, effect size and frequency are major factors in determining the power to detect associations. In the absence of any selection-driven relationship between effect size and frequency, lower frequency GWAS alleles tend to have bigger effects (Park et al 2011 PNAS, “Distribution of allele frequencies …”). Allele frequency is also a major determinant of Fst — low frequency alleles are more constrained in terms of the values of Fst that they can obtain (see the Jakobsson citation above). This could result in spurious correlations between effect sizes and Fst, or any other statistic whose values are partially determined by allele frequency.

Minor comments:

Methods: Can the authors clarify how they have performed the LD matching? I could not follow the details in the manuscript. It seems like they have pruned on r^2, but this is a pairwise metric and it was unclear to me how they extend it to a genomic region. Do they match the LD spectrum? Did they consider other methods to match LD, such as matching on LD score?

Methods: How do the authors ensure that there are enough matched genomic background regions to provide a good null? In particular, when using 500kb windows, there are only ~6000 such windows in the genome. If a GWAS has several hundred independent hits, are there enough windows remaining that match on LD and frequency such that the different random background samples are actually independent, and not formed repeatedly from a small subset of windows? The tests could be anti-conservative if the set of windows making of the background are highly correlated across each of the 5,000 samples, resulting in correlations between each set that do not reflect the true variance.

Abstract: "was the most dominant", this seems ambiguous please clarify.

Abstract: is “negative enrichment” the same as “depletion”? Perhaps depletion would be clearer.

The first paragraph makes it sound like selection is known to be the most important evolutionary process affecting human differentiation. Drift is also undoubtedly very important. I think any group of population geneticists could argue unproductively over the specifics for hours, but the intro should make it clear that drift has a substantial role and exactly how much differentiation is fitness-related and how much is neutral is not known.

Line 94: "Despite these complications..." This makes it sound like these studies have gotten around the stratification issue to find robust selection evidence. I agree that some polygenic selection findings are more robust than others, but I'm not aware of any that completely remove any possible confounding. If the authors disagree they should provide a reason, as I do not think this is clear (at least to me).

Line 101-102: It’s true that many previous papers have been motivated by understanding the processes that could shape traits, but many have been clear that we cannot distinguish between selection on the traits and selection on trait variation.

Line 424: “A second major difference is that we do not directly consider effect size or direction inferred 425 from GWAS”, I agree with the author’s point in this section that this can avoid some issues with stratification. However, alleles near the threshold for detection (alleles with small effect sizes that barely pass the threshold for significance) may still be stratified because they are more likely to be false positives. Some previous methods have also used only the sign of the effect size and not the magnitude (e.g. Turchin et al).

Thanks for the opportunity to review your work!

Sincerely,

Lawrence Uricchio

**Have all data underlying the figures and results presented in the manuscript been provided?**

Reviewer #1: Yes

Reviewer #2: Yes

Reviewer #3: Yes

PLOS authors have the option to publish the peer review history of their article (what does this mean?). If published, this will include your full peer review and any attached files.

Reviewer #1: No

Reviewer #2: No

Reviewer #3: **Yes: **Lawrence H. Uricchio

---

## [Editor Report · Decision Letter 1]

21 Oct 2022

Dear Dr Rokas,

We are pleased to inform you that your manuscript entitled "Mosaic patterns of selection in genomic regions associated with diverse human traits" has been editorially accepted for publication in PLOS Genetics. Congratulations!

Yours sincerely,

Justin C. Fay

Academic Editor

PLOS Genetics

Bret Payseur

Section Editor

PLOS Genetics

Comments from the reviewers (if applicable):

**Data Deposition**

http://datadryad.org/submit?journalID=pgenetics&manu=PGENETICS-D-22-00670R1

**Press Queries**

---

## [Editor Report · Acceptance letter]

31 Oct 2022

PGENETICS-D-22-00670R1 

Mosaic patterns of selection in genomic regions associated with diverse human traits 

Dear Dr Rokas, 

We are pleased to inform you that your manuscript entitled "Mosaic patterns of selection in genomic regions associated with diverse human traits" has been formally accepted for publication in PLOS Genetics! Your manuscript is now with our production department and you will be notified of the publication date in due course.

With kind regards,

Zsofia Freund

PLOS Genetics

On behalf of:
